# Patterns of Sequelae in Women with a History of Localized Breast Cancer: Results from the French VICAN Survey

**DOI:** 10.3390/cancers13051161

**Published:** 2021-03-08

**Authors:** Lidia Delrieu, Liacine Bouaoun, Douae El Fatouhi, Elise Dumas, Anne-Deborah Bouhnik, Hugo Noelle, Emmanuelle Jacquet, Anne-Sophie Hamy, Florence Coussy, Fabien Reyal, Pierre-Etienne Heudel, Marc-Karim Bendiane, Baptiste Fournier, Mauricette Michallet, Béatrice Fervers, Guy Fagherazzi, Olivia Pérol

**Affiliations:** 1Department Prevention, Cancer, Environment, Léon Bérard Cancer Center, 69008 Lyon, France; lidia.delrieu@lyon.unicancer.fr (L.D.); hugo.noelle@lyon.unicancer.fr (H.N.); baptiste.fournier@lyon.unicancer.fr (B.F); beatrice.fervers@lyon.unicancer.fr (B.F.); 2Residual Tumor & Response to Treatment Laboratory, RT2Lab, Translational Research Department, INSERM, U932 Immunity and Cancer, Institut Curie, Paris University, 75005 Paris, France; elise.dumas@curie.fr (E.D.); anne-sophie.hamy-petit@curie.fr (A.-S.H.); florence.coussy@curie.fr (F.C.); fabien.reyal@curie.fr (F.R.); 3International Agency for Research on Cancer, 69372 Lyon, France; bouaounl@iarc.fr; 4Center of Research in Epidemiology and Population Health, UMR 1018 Inserm, Institut Gustave Roussy, Paris-Sud Paris-Saclay University, 94807 Villejuif, France; douae.el-fatouhi@gustaveroussy.fr (D.E.F.); guy.fagherazzi@lih.lu (G.F.); 5MINES ParisTech, PSL Research University, CBIO-Centre for Computational Biology, 75006 Paris, France; 6INSERM, IRD, SESSTIM, Economics & Social Sciences Applied to Health & Analysis of Medical Information, Aix Marseille University, 13007 Marseille, France; anne-deborah.bouhnik@inserm.fr (A.-D.B.); marc-karim.bendiane@inserm.fr (M.-K.B.); 7Oncology and Blood Diseases Department, University Hospital Center, Joseph Fourier University, CEDEX 9, 38043 Grenoble, France; ejacquet1@chu-grenoble.fr; 8Department of Medical Oncology, Institut Curie, 75005 Paris, France; 9Department of Surgical Oncology, Institut Curie, University Paris, 75005 Paris, France; 10Department of Medical Oncology, Léon Bérard Cancer Center, 69008 Lyon, France; pierreetienne.heudel@lyon.unicancer.fr (P.-E.H.); mauricette.michallet@lyon.unicancer.fr (M.M.); 11Cancer Research Center of Lyon, INSERM UA8, Léon Bérard Cancer Center, 69008 Lyon, France; 12Department of Population Health, Luxembourg Institute of Health (LIH), 1445 Strassen, Luxembourg

**Keywords:** sequelae, breast cancer, prevention, VICAN

## Abstract

**Simple Summary:**

Breast cancer induces sequelae even years after diagnosis, but little is currently known about long-term sequelae patterns. This study aimed to (1) assess the evolution of the main sequelae and treatment two and five years after diagnosis in women with early-stage breast cancer, (2) explore patterns of sequelae associated with given sociodemographic, clinical, and lifestyle factors. Our results show that six main classes of sequelae were identified and remain constant over time except for fatigue and cognitive sequelae. The latent class analysis identified two main classes of sequelae (functional and esthetic patterns) and we have highlighted that different risk factors such as treatment, sociodemographic level, and physical activity level were associated with an increased risk of long-term sequelae.

**Abstract:**

Breast cancer (BC) remains complex for women both physically and psychologically. The objectives of this study were to (1) assess the evolution of the main sequelae and treatment two and five years after diagnosis in women with early-stage breast cancer, (2) explore patterns of sequelae associated with given sociodemographic, clinical, and lifestyle factors. The current analysis was based on 654 localized BC patients enrolled in the French nationwide longitudinal survey “vie après cancer” VICAN (January–June 2010). Information about study participants was collected at enrollment, two and five years after diagnosis. Changes over time of the main sequelae were analyzed and latent class analysis was performed to identify patterns of sequelae related to BC five years after diagnosis. The mean age (±SD) of study participants at inclusion was 49.7 (±10.5) years old. Six main classes of sequelae were identified two years and five years post-diagnosis (functional, pain, esthetic, fatigue, psychological, and gynecological). A significant decrease was observed for fatigue (*p* = 0.03) and an increase in cognitive sequelae was reported (*p* = 0.03). Two latent classes were identified—functional and esthetic patterns. Substantial sequelae remain up to five years after BC diagnosis. Changes in patient care pathways are needed to identify BC patients at a high risk.

## 1. Introduction

Breast cancer (BC) is the leading cause of cancer among women worldwide with about 2.3 million new cases in 2020 [1]. In France in 2020, there were about 60,000 new BC cases estimated in women [1] and thanks to major therapeutic advances, the overall 5-year survival rate of BC has increased from 80% for women diagnosed in 1989–1993 to 87% for those diagnosed in 2005–2010 [2].

The management of BC remains complex from a psychological point of view, as well as physically, mainly due to the high number of routine treatments available (chemotherapy, radiotherapy, hormonotherapy, and/or surgery including reconstructive surgery). Hence, many side effects and sequelae may persist long after treatment. Many patients report subsequent fatigue [3,4,5,6], poor quality of life [7], pain [8,9], cognitive impairment [10,11], functional disability associated with lymphoedema [12,13,14,15], heart [16,17,18,19,20], respiratory [16], femininity [21,22], and sexual dysfunctions [23] during or immediately after treatment, but very little is known about long-term sequelae of BC. Moreover, people who suffered from cancer remain in poorer health than the general population [24,25,26]. In addition, cancer patients have a higher risk of developing a second primary cancer [27,28] and the accumulation of sequelae due to cancer can lead to premature death [29,30,31]. However, a subset of sequelae are modifiable and could be reduced through lifestyle changes such as physical activity and diet, and medical follow-up [32,33].

The “vie après cancer (VICAN)” French national survey was conducted two years (VICAN2) and five years (VICAN5) after a cancer diagnosis and aimed at investigating the long-term impact of cancer on patients’ lives (health, sequelae, everyday life difficulties, return to work, etc.) [34,35]. 

The objectives of the current study were to (1) assess the evolution of the main sequelae related to cancer and its treatments two years and five years after diagnosis in women with early-stage breast cancer, (2) identify and characterize subpopulations of BC patients with similar patterns of sequelae. 

## 2. Materials and Methods

### 2.1. Study Population

The VICAN survey is a longitudinal survey among French adults aged 20–85 and diagnosed with cancer between January and June 2010 with data collected at two time points. The first data wave was conducted in 2012, two years after diagnosis, and included 4349 participants (VICAN2 overall response rate 43.7%). The second wave was undertaken in 2015, five years after diagnosis, and involved 2165 participants (VICAN5) which represents a total of 4174 patients among whom 2009 participants had already participated at VICAN2, with an additional sample of 2165 who did not respond to VICAN2 (Figure 1). The survey was designed to monitor health conditions two and five years since diagnosis with any type of cancer. Survey procedures and methods have been previously published [34]. Briefly, patients diagnosed with a cancer among twelve cancer types (breast, lung, colorectal, prostate, upper aerodigestive tract, bladder, kidney, cervical, endometrial, thyroid, non-Hodgkin lymphoma, melanoma) were identified through electronic health care system records derived from one of the main French health insurance schemes—Caisse Nationale de l’Assurance Maladie des Travailleurs Salariés (CNAMTS) for salaried workers, Régime Social des Indépendants (RSI) for self-employed workers, and Mutuelle Sociale Agricole (MSA) for farmworkers. Data were collected through questionnaires administered by telephone surveys two and five years after cancer diagnosis, while administrative and medical data were extracted from the national medico-administrative database on reimbursement data for healthcare costs and hospital discharge records (Système National des Données de Santé, SNDS).

In this current study, we focused on women diagnosed with an early BC who responded to both two- and five-year follow-up forms, leading to a total of 654 out of 4349 participants. 

### 2.2. Data Collection

Data collected by the survey at diagnosis, two years (VICAN2) and five years (VICAN5) after female BC diagnosis, included the following information:

#### 2.2.1. Data on Spontaneous Sequelae

Spontaneous sequelae were explicitly reported by patients with open-ended questions for both VICAN2 and VICAN5 surveys. For the study purpose, all answers to the questions mentioned above were analyzed individually according to keywords and expressions to identify the main theme(s) (Appendix A). Twenty main sequelae were identified and reported similarly to sequelae found in the literature [36]: functional (total functional disorders, arm-related functional disorders, other functional disorders), pain, esthetic, fatigue, psychological, gynecological, weight, skin, cognitive, respiratory, vision, sleep, digestive, sexual, cardiac, hearing, financial, dental, urinary, and others. All sequelae were coded in binary (yes/no). Two other closed-ended questions were asked concerning sequelae. For the presence and magnitude of sequelae, patients had to answer to the question “Generally speaking, do you keep any sequelae due to your illness?” in a five-point Likert scale 1—“yes, and they are very important”, 2—“yes, and they are important”, 3—“yes, but they are moderate”, 4—“yes, but they are very moderate” and 5—“no, I do not have any sequelae”. Patients were asked if they received specific treatment for their sequelae (yes/no). Women could have several types of sequelae. The variable “total number of reported sequelae” was created by summing all the binary variables according to the 20 main reported sequelae. 

Women were allocated to the group without sequelae if they (i) did not report explicitly sequelae in the open-ended questions, (ii) had not answered the question about the magnitude of the sequelae, and (iii) had not answered the question about the treatment of the sequelae.

#### 2.2.2. Socio-Demographic 

Socio-demographic information included age, level of education (<high school vs. ≥high school), marital status (married/partners vs. single/divorced/separated), occupational status ((i) employed and performing traded; (ii) employed in managerial occupations; or (iii) unemployed), and employment status (in employment including being in short-term sick leave vs. another situation).

#### 2.2.3. Medical Characteristics 

Data were extracted from the medico-administrative databases (Système national d’information inter-régimes de l’Assurance maladie, SNIIR-AM). Information on treatments: chemotherapy (yes/no), hormone therapy (yes and still taking the treatment/yes no longer taking the treatment/no), radiotherapy (yes/no) and reconstructive surgery (yes/no) were collected and extracted for the present analysis. 

#### 2.2.4. Self-Rated Global Health

Self-perceived health status was evaluated using a question from the Minimum European Health Module with a five-point Likert scale with items 1—“very good”, 2—“good”, 3—“quite good”, 4—“bad”, and 5—“very bad” [37].

#### 2.2.5. Comorbidity

Comorbidity score was assessed by the Individual Chronic Condition score using a validated pharmacy-based measure at which cancer was removed, using medico-administrative data from 2015 [38]. This score was calculated as a weighted average number of chronic diseases identified in an individual over the previous year by the hospitalization or mortality rate of the disease.

#### 2.2.6. Neuropathic Pain

Neuropathic or nociceptive pain score was measured using the validated the Douleur Neuropathique 4 (DN4) questionnaire [39]. It consists of ten items. The first seven items were dedicated to the presence of the symptoms of pain (burning, painful cold, electric shock, tingling, pins and needles, numbness, and itching). The remaining three items were completed after a clinical examination (hypoesthesia to touch, hypoesthesia to pinprick, pain caused or increased by brushing). The items of the DN4 questionnaire are scored based on a yes (1 point)/no (0 points) answer. A total score of 4 out of 10 or more suggests neuropathic pain. 

#### 2.2.7. Fatigue

Fatigue was measured according to the validated European Organization for Research and Treatment of Cancer Quality of Life Questionnaire (EORTC QLQ) core 30 [40,41] using the fatigue subscale. This subscale comprises three fatigue-related items over the previous week: “Did you need to rest?”, “Have you felt weak?”, “Were you tired?”. A four-point Likert scale was used “1 = not at all”, “2 = a little”, “3 = quite a bit”, “4 = very much”. The score was linearly transformed to a 0–100 scale and the higher the score, the greater the fatigue. 

#### 2.2.8. Tobacco use and Alcohol Consumption

Tobacco use and alcohol consumption were categorized in smokers vs. non-smokers and drinkers vs. non-drinkers, respectively. Smokers were defined as those who responded “smoking daily or less often” and non-smokers as those who responded “not smoking”. Drinkers were defined as those who responded “drink once a month or more” and non-drinkers as those who responded “not to drink”.

#### 2.2.9. Anthropometrics

Anthropometrics were assessed using body mass index (BMI) calculated as a ratio of self-reported body weight (in kg) to body height (in meters) squared.

#### 2.2.10. Physical Activity

Participants had to answer to the question “Since the diagnosis of your illness, has your physical activity changed?” “1 = yes, I do more than before”, “2 = yes, I do less than before”, “3 = yes, I have completely stopped”, “5 = no”.

### 2.3. Statistical Analyses

Participants’ characteristics were described using means and standard deviations (SDs) for quantitative data and using frequencies and percentages for qualitative data.

The difference in numbers of sequelae between two and five years after diagnosis were investigated using a Wilcoxon signed-rank test and differences in presence or absence of sequelae were evaluated using the McNemar’s test. Out of the 20 sequelae identified, we only considered the main sequelae defined by the ones present in at least 5% at 2-year post-diagnosis.

A latent class analysis (LCA) was performed to identify subgroups (latent classes) of BC patients with similar patterns of sequelae five years after diagnosis [42,43]. The LCA model has proven useful for discrete and dichotomous variables to identify subpopulations sharing similar item response patterns. The LCA included the main six binary (no/yes) sequelae variables identified five years after diagnosis: functioning, pain, esthetic, fatigue, psychological, and gynecological sequelae. The choice of the optimal number of latent classes, k, was based on the adjusted Bayesian Information Criterion (aBIC) as a trade-off between parsimony and model fit. We implemented LCA models with 2–8 latent classes and selected the model with the lowest aBIC criteria corresponding to the two (k = 2) classes model. After selecting the best model, the probability of each latent class membership, the conditional probability of having each sequela in each latent class, and the classification of patients based on their most likely latent-class membership were finally reported. Conditional probabilities denoted the probabilities of the sequelae being present, conditional on specific class membership. For each patient, we obtained posterior probabilities of belonging to the different latent classes of the LCA model and which were then classified into the class for which they have the highest posterior membership probability. Latent classes were also described and characterized by sequelae profiles.

Univariate logistic regression models were performed to investigate the associations between the latent class membership and socio-demographic at inclusion, lifestyle at VICAN2, and clinical variables at VICAN5. Variables included age (<50 vs. ≥50 years), occupational status (trades, managerial occupations, no employment), education level (<high school or ≥high school), marital status (single vs. partners), physical activity level (stop, decrease, stable, or increase), BMI (underweight: <18.5 kg/m^2^, normal: <25 kg/m^2^, overweight: 25–30 kg/m^2^, and obese: >30 kg/m^2^), smoking and alcohol consumption (yes vs. no), chemotherapy (yes vs. no), radiotherapy (yes vs. no), hormone therapy (still in treatment, no longer in treatment, no), reconstructive surgery (yes vs. no), comorbidity, pain, and fatigue scores as continuous variables. Thereafter, a multivariate analysis was carried out using a subset of these variables known or considered as variables impacting the occurrence of sequelae after being tested in univariate models. Odds ratios (ORs) with 95% confidence intervals (CIs) of both univariate and multivariate analyses were calculated and reported. All statistical analyses were performed using R version 3.6.1 and a two-sided *p* value < 0.05 was considered as statistically significant.

## 3. Results

### 3.1. Participants’ Characteristics

The participants’ descriptive characteristics at diagnosis are presented in Table 1. The mean age (±SD) was 49.8 (±10.5) years old. Women were, in majority, married (70%), and with children (88%) (data not shown). Most of the patients had a high education (higher than or equal to high school; 56%) and the majority of participants were in employment or on sick leave at diagnosis (72%). Treatments that patients received up to five years since diagnosis were radiotherapy (81%), chemotherapy (59%), and hormone therapy (72%), and they still received hormone therapy five years after diagnosis (42%) (Table 2).

### 3.2. Main Sequelae of BC and Changer over Time

The mean BMI was 24.7 kg/m^2^ and 58% had a normal weight two years post-diagnosis (Table 2). Regarding physical activity, half of the patients maintained or increased their physical activity two years after diagnosis while this proportion remained stable five years after (VICAN5) diagnosis (49%). Participants remained at a moderate fatigue level at two years (50.3 ± 27.6) and five years post-diagnosis (48.6 ± 26.9).

Changes in sequelae two years and five years after diagnosis are presented in Table 2 and Figure 2. Two years after diagnosis, 471 (72%) participants still reported suffering from sequelae including 31% intense and important; five years after diagnosis there were almost as many patients suffering from sequelae (*n* = 448) and many of them (46%) did not receive any specific treatment for the management of these sequelae (Table 2). Five years after diagnosis (VICAN5), the number of sequelae remained stable (average: 1.6, *p* = 0.59). We observed a significant decrease in fatigue (16% vs. 12%, *p* = 0.03) as well as an increase in cognitive sequelae (2% vs. 4%, *p* = 0.03) between two and five years after diagnosis while all other sequelae remain stable (Figure 2). Participants who suffered from pain sequelae mostly have associated functional sequelae, while women with psychological sequelae often experience great fatigue and esthetic sequelae (Figure 3).

### 3.3. Latent-Class Analysis of Sequelae Patterns

Latent class membership and conditional response probabilities within each latent class of the two latent class model at five years (VICAN5) are given in Appendix A. About 52% of participants were categorized in class 1 and 48% in class 2. Class 1 was characterized by patients with esthetic sequelae while class 2 by functional and pain sequelae among the six investigated. These two latent classes are from now on referred to as “esthetic pattern” and “functional pattern” classes, respectively.

The characteristics of the patients according to their class are shown in Table 3. Women belonging to the “esthetic pattern” class are older women who received less hormonotherapy and chemotherapy, and who had maintained or increased their level of physical activity since diagnosis as compared to the “functional pattern”.

In logistic regression models, class 1 “esthetic pattern” was chosen as the reference class when presenting ORs as it corresponded to the class having the lowest probability for getting all sequelae. Univariate ORs for class membership compared to the reference class are presented in Table 4. Results showed that treatment by chemotherapy (OR = 1.63, 95% CI = 1.19–2.56) and current hormone therapy (1.53 [1.07–2.19]), fatigue and pain scores (1.02 [1.01–1.03] and 1.14 [1.04–1.26] respectively) and being married or with a partner (1.42 [1.01–1.09]) were associated with an increased risk of being in the “functional pattern” class. Moreover, reducing physical activity level is also associated with an increased risk of belonging to the “functional pattern” class (1.77 [1.23–2.56]). In contrast, being over 50 years of age (0.68 [0.49–0.95]) was associated with a decreased risk of belonging to the same class.

Results from multivariate analyses (Figure 4) showed mainly similar findings with those from univariate analyses with a few exceptions: performing trades (0.65 [0.44–0.95]) decreased the risk of belonging to the “functional” class five years after the BC diagnosis. For the class 2 “functional pattern”, while age, chemotherapy, hormone therapy, and marital status were no longer associated with an increased risk of belonging to this class, the comorbidity score at diagnosis (1.72 [1.04–2.85]) was significantly associated with a higher risk of being in that class. Significant associations are presented in bold.

## 4. Discussion

### 4.1. Principal Findings

While the survival of female BC patients after diagnosis and treatment has improved, this study revealed that they suffered from sequelae five years after the diagnosis of BC yet. To the best of our knowledge, this is the first study to identify subgroups of sequelae associated with their risk factors at five years since diagnosis in patients with early BC using the LCA model. The LCA identified two main classes of sequelae (functional and esthetic patterns) and we have highlighted that different risk factors such as treatment (chemotherapy, hormone therapy, and surgery), sociodemographic level, and physical activity level were associated with an increased risk of long-term sequelae.

### 4.2. BC Sequelae

BC and its treatments disrupt patients’ lives and lead to significant long-term consequences. Most BC patients have undergone important physical mutilation and must follow hormone therapy treatments for five years. This study showed that more than 70% of patients had sequelae that lasted at least five years after diagnosis, which is consistent with the results in a previous study reporting that about 60% of women have at least one or more sequelae six years after BC diagnosis [44]. The study goal was to describe and understand the prevalence of breast cancer treatment sequelae among 287 Australian breast cancer women. Despite the descriptive aspect of the study with a smaller workforce, this study reinforces the need to better identify women’s sequelae and to provide appropriate care. Our study identified 20 sequelae grouped into six sequelae most strongly represented in this study population (functional, pain, esthetic, fatigue, psychological, and gynecological sequelae) [45]. Despite the reconstruction offered to 25%, esthetic and psychological sequelae persisted five years after the diagnosis and altered both body image and activities of daily living. Of note, organic sequelae such as cardiac dysfunctions were poorly represented despite anthracycline and trastuzumab treatments are known to induce cardiac complications in BC because of lack of a survey with exploratory parameters by biological, echography, or more sophisticated tests. In previous studies, chemotherapy and hormone therapy were associated with functional and fatigue sequelae [4,46,47,48] and patients who experienced such sequelae were mostly younger. Hormone therapy has improved the survival of patients with breast cancer but induced many sequelae that impact quality of life, social interaction, and the everyday life [49,50,51]. Patients are faced with menopause and its associated symptoms (hot flashes, sleeping difficulties, gynecological complications) [52,53]. Sequelae of hormone therapy have been often underestimated whereas the effects can compromise its long-term adherence [49]. A study with 2353 women with breast cancer showed that one-third of women did not feel supported by the medical team to overcome sequelae [51]. These results underline the need for early detection of sequelae and better support for patients.

### 4.3. Supportive Care for Patients with Cancer

Recently, there has been an awareness to offer global care to the patient including supportive care to reduce physical and psychological effects of BC and its treatments. Many efforts remain to be made in terms of prevention or early detection with specific adapted interventions [54]. Therapeutic patient education programs complementary to traditional healthcare have been increasingly offered to patients [55,56]. Indeed, patient empowerment is defined as an effective approach to help people to gain control over their lives, to increase their capacity to act, to manage their illness and side effects of treatments, and to become autonomous [57,58]. While today, supportive care is mainly offered from the moment of diagnosis, there is currently a gap in patient care after treatments with inequalities between institutions. Patients often have one follow-up appointment per year mainly focused on a potential recurrence and little on the patients’ physical and psychological sequelae. Patients often feel helpless and alone to return to a normal life and there is a real need for patients to have a close global long-term follow-up as 45% of them report not having any treatment for their sequelae.

### 4.4. Benefits of Physical Activity

Physical activity is considered as a supportive care and has shown many physical, biological, psychological, and clinical benefits for patients with BC [59,60,61]. Nevertheless, most of the studies were carried out on cross-sectional, retrospective, or short-term intervention study, which does not allow us to conclude on the medium- and long-term effects of physical activity in preventing sequelae over time. Despite the major interest in maintaining or increasing physical activity from diagnosis, we note in this survey that about half of the patients decrease or stop practicing physical activity both at two years and five years after diagnosis. While the level of physical activity tends to decrease at diagnosis [12], it seems important to provide intervention programs to BC patients from diagnosis to maintain physical activity over time and, in turn, to prevent the risk of sequelae. A prospective study conducted in 201 Canadian BC survivors is currently in progress, to study the longitudinal associations between lifestyle behavior changes and physical health and sequelae during four years [62]. However, to date, no physical activity intervention with objective measurement of physical activity has been carried out to study the associations between physical activity level and the risk of sequelae. Only one systemic review in breast cancer patients receiving hormone therapy concluded that aerobic plus resistance training had positive effects on cardiorespiratory fitness and pain, but further studies are needed to confirm the associations [61].

### 4.5. Individualized Care for Patients with Cancer

Thus, it is now necessary to offer cancer patients individualized care from the time of diagnosis to reduce medium- and long-term sequelae. With the implementation of the VICAN study, there has been an awareness of the importance of screening and predicting sequelae [63,64] and new programs tend to be implemented in cancer centers to offer a new care path. At diagnosis, patients are not able to anticipate future sequelae and many believe that sequelae are normal and are part of the cancer disease and treatment. Thus, to optimize patient care, it would be necessary, on the one hand, to identify patients at risk of sequelae development and to guide patients to offer them appropriate and personalized care. The use of questionnaires at diagnosis or at the end of treatment to assess the level of fatigue, comorbidity, pain, physical activity, coupled with clinical data related to treatments received, could help identify patients at risk of sequelae development, mainly esthetic or functional pattern. However, some women may suffer from different types of sequelae at the same time. Then, having simple but valid tools to obtain objective data, particularly in terms of organic sequelae, would be a considerable step forward.

### 4.6. Strengths and Weaknesses

The strength of the study is that we identified sequalae patterns within the BC patients and examined its associations with their clinical, socio-demographic, and lifestyle factors, which have never been investigated before. The longitudinal design with a large sample size allows monitoring the medium and long-term sequelae.

However, there were several limitations in this study. First of all, the VICAN survey was conducted among a particular population since women were relatively young at diagnosis (49.8 years) and highly educated (56% higher than or equal to high school), which would not be representative of the population with a median age at diagnosis of cancer, about 63 years in France [2]. The youthful population of the survey can be explained by the fact that the survey was initially carried out to study the return to work of cancer patients which implies recruiting patients before the age of retirement. Secondly, the item sequelae were transcribed in free text and it was necessary to manually recode all the information into different categories, which implies a certain amount of subjectivity. Third, some variables used for the analyses were collected in subjective ways and collected at different periods (at inclusion or at two years or at five years). For instance, changes in physical activity levels of patients since diagnosis were not measured by validated questionnaires or by objective tools such as accelerometers to objectively quantify their physical activity level. Fourth, regarding data quality, some data (or variables) were not standardized using different collecting methods between VICAN2 and VICAN5 surveys. Fifth, some information related to BC sequelae was missing, such as type of chemotherapy and type of surgery. Some chemotherapy treatments, particularly anthracycline, can induce cardiac sequelae [65,66], neuropathic pains due to taxane [67,68]. The type of surgery (mastectomy, tumorectomy) can also induce different functional, esthetic, and psychological sequelae, which have probably an important impact on all variables examined in this study. The LCA model has a methodological limitation when determining the optimal number of classes regardless of the criteria used (aBIC, BIC, AIC) that might be improved by a larger sample size. We tested several different classes to propose the best statistical model.

## 5. Conclusions

Our study showed that significant sequelae have remained five years after diagnosis. This study suggests an association between a decrease in physical activity and sequelae at five years after BC diagnosis. Further studies measuring accurately physical activity are required to confirm this finding. Moreover, better management and/or screening for pain and fatigue could also help to prevent future sequelae. Changes in patient care pathways are needed to identify patients at risk and/or screen sequelae to provide personalized and appropriate management including supportive care.

## Figures and Tables

**Figure 1 cancers-13-01161-f001:**
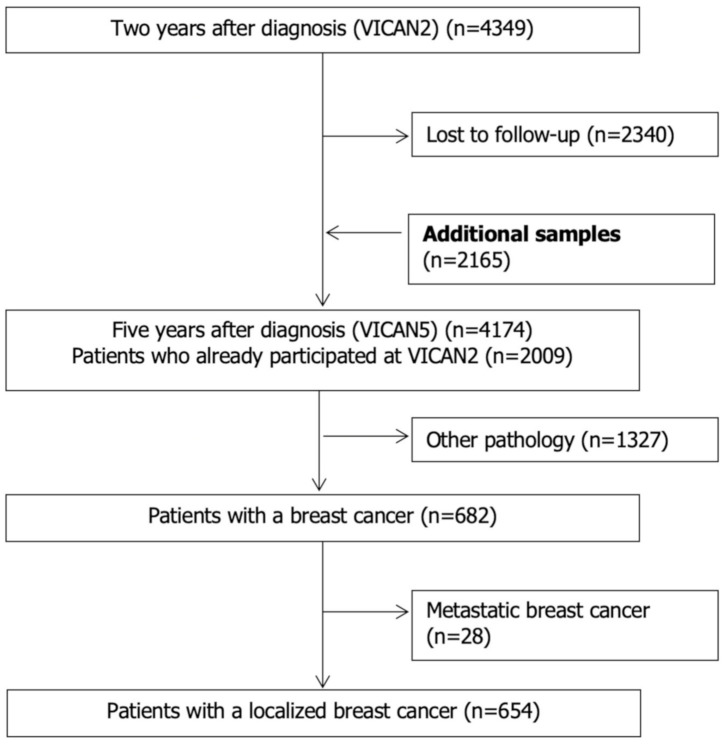
Flow chart of the study population, “vie après cancer” (VICAN) (*n*= 654).

**Figure 2 cancers-13-01161-f002:**
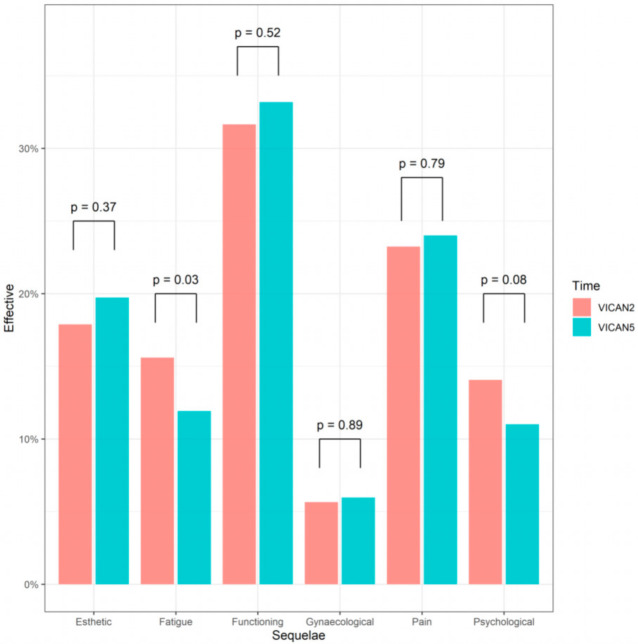
Evolution of sequelae between two years (VICAN2) and five years (VICAN5) after diagnosis of the six main sequelae (*n*= 654).

**Figure 3 cancers-13-01161-f003:**
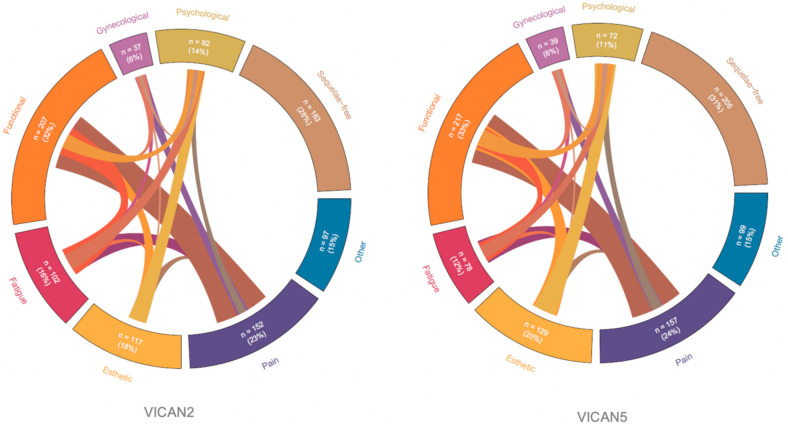
Distribution of the six main sequelae (*n* = 654). Figure depicts the distribution and co-occurrence of sequelae among patients at two years (VICAN2) (left) and five years (VICAN5) (right). The strength of the association between two sequelae is reflected by the edge width of the ribbon.

**Figure 4 cancers-13-01161-f004:**
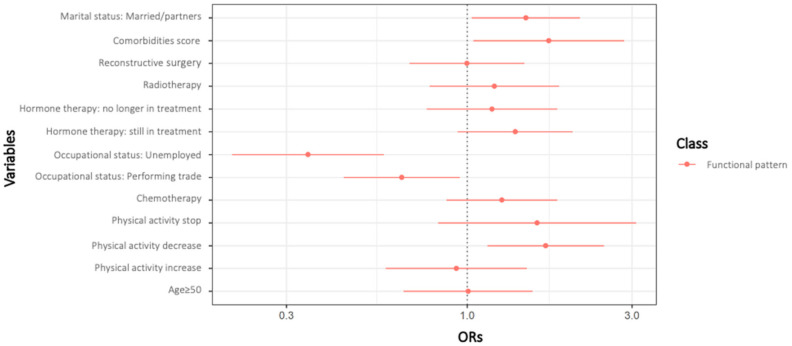
Adjusted multinomial logistic regression analysis (*n*= 654).

**Table 1 cancers-13-01161-t001:** Characteristics of the study population (*n* = 654) ^a^ Missing data = 1.

Characteristics	Mean (SD)	*n* (%)
**Age at diagnosis** (years)	49.8 (10.5)	
**Age at diagnosis** (years)		
<50		438 (67)
>50		216 (33)
**Marital status ** ^a^		
Married/Partners		458 (70)
Single/Divorced/Separated		195 (30)
**Education**		
<High school		286 (44)
≥High school		368 (56)
**Occupational status**		
Employed		
Performing trades (farmers, craftsmen, laborers, employees)		271 (42)
Managerial occupations (executives and senior managers, heads of companies, intermediate professions)		198 (30)
Unemployed		185 (28)
Employment status		
In employment (or short-term sick leave / leave)		469 (72)
Other situations		185 (28)

^a^ Missing data at VICAN2 = 3 and missing data at VICAN5 = 8.

**Table 2 cancers-13-01161-t002:** Anthropometric measures, lifestyle factors, clinical data, and changes over time in sequelae at two and five years after diagnosis (*n*= 654).

Characteristics	Two Years (VICAN2)		Five Years (VICAN5)	
	Mean (SD)	*n* (%)	Mean (SD)	*n* (%)
**LIFESTYLE**				
**Anthropometry**				
BMI (kg/m^2^)	24.6 (5.6)		-	
Underweight (<18.5 kg/m^2^)		26 (4)		-
Normal weight (<25 kg/m^2^)		382 (58)		-
Overweight (25–30 kg/m^2^)		170 (26)		-
Obese (>30 kg/m^2^)		76 (12)		-
**Physical activity evolution since diagnosis** ^a^				
Stop		48 (7)		38 (6)
Decrease		275 (42)		295 (46)
Stable		203 (31)		204 (32)
Increase		125 (19)		109 (17)
**Smoking (yes)**		128 (20)		135 (21)
**Alcohol consumption** (**yes**)		492 (75)		506 (77)
**Treatment**				
Chemotherapy since diagnosis		382 (58)		387 (59)
Radiotherapy since diagnosis		532 (81)		532 (81)
Hormonotherapy since diagnosis				
Yes, and I’m still taking this treatment		442 (68)		275 (42)
Yes, and I’m no longer taking this treatment		26 (4)		163(25)
No		186 (28)		215 (33)
Reconstructive surgery		NA		164 (25)
**Pain score** (**DN4**) ^b^	2.4 (1.8) ^c^		2.2 (1.8) ^c^	
**Fatigue score** (**EORTC**)	50.3 (27.6)		48.6 (26.9)	
**Comorbidity score**	0.5 (0.4)		0.6 (0.4)	
**Self-rated global health** ^c^				
Very good		127 (19)		34 (5)
Good		252 (39)		80 (12)
Quite good		236 (36)		422 (65)
Bad		37 (6)		86 (13)
Very bad		0		25 (4)
**Presence of sequelae**		471 (72)		448 (69)
**Number of sequelae** (mean, SD)		1.6 (1.4)		1.6 (1.5)
**Presence and magnitude of sequelae following the management of your disease** ^c^				
Intense		53 (8)		34 (5)
Important		146 (23)		117 (18)
Moderate		182 (28)		204 (31)
Very moderate		92 (14)		100 (15)
No sequelae		174 (27)		195 (30)
**Specific treatment for sequelae** ^d^				
Yes		-		448 (69)
No		-		1.6 (1.5)

^a^ Missing data at VICAN2=3 and missing data at VICAN5 = 8; ^b^ Missing data at VICAN2 and VICAN5 = 111; ^c^ Missing data at VICAN2 = 2 and missing data at VICAN5 = 7; ^d^ Missing data at VICAN5 = 200.

**Table 3 cancers-13-01161-t003:** Characteristics of the two latent classes model among patients with breast cancer (BC) (*n* = 654).

	Class 1 (%)	Class 2 (%)
Patterns:	Esthetic Pattern	Functional Pattern
**N**	340	314
**Sequelae**		
Functioning	6.8	61.8
Pain	0	50.0
Esthetic	31.2	7.3
Fatigue	0	24.8
Psychological	10.9	11.1
Gynaecological	0	12.4
**SOCIODEMOGRAPHIC AT INCLUSION**		
**Age at diagnosis** (years) mean (SD)	51.2 (11.3)	48.4 (9.5)
<50	63.0	71.3
≥50	37.0	28.7
**Education**		
<High school	47.6	39.5
≥High school	52.4	60.5
**Marital status**		
Married/partners	66.7	73.9
Single/Divorced/Separated	33.3	26.1
**Occupational status**		
Performing trades (farmers, craftsmen, labourers, employees)	39.7	43.3
Managerial occupations (executives and senior managers, heads of companies, intermediate professions)	24.7	36.3
Not in employment at the time of diagnosis	35.6	20.4
**LIFESTYLE AT VICAN2**		
**BMI** (kg/m^2^) mean (SD)	24.5 (6.2)	24.6 (4.9)
**BMI** (kg/m^2^)		
Underweight (<18.5 kg/m^2^)	2.9	5.1
Normal weight (<25 kg/m^2^)	61.2	55.4
Overweight (25–30 kg/m^2^)	25.0	27.1
Obese (>30 kg/m^2^)	10.9	12.4
**Physical activity evolution since diagnosis**		
Stop	6.5	8.3
Decrease	37.0	48.0
Stable	35.8	26.2
Increase	20.7	17.5
**Smoking** (**yes**)	17.9	21.3
**Alcohol consumption** (**yes**)	76.8	73.6
**TREATMENTS AT VICAN5**		
**Radiotherapy**		
Yes	79.4	83.4
No	20.6	16.6
**Chemotherapy**		
Yes	53.5	65.3
No	46.5	34.7
**Hormonotherapy since diagnosis**		
Yes, and I’m still taking this treatment	37.5	47.1
Yes, and I’m no longer taking this treatment	26.5	23.2
No	36.0	29.7
**Reconstructive surgery**		
Yes	76.1	73.6
No	23.9	26.4
**Fatigue score** (mean, SD)	43.0 (27.8)	58.2 (25.2)
**Comorbidities score** (mean, SD)	0.7 (0.4)	0.7 (0.3)

**Table 4 cancers-13-01161-t004:** Univariate logistic regression odds ratios of potential risk factors for latent class membership (compared to esthetic pattern) of BC women from the VICAN study (*n* = 654).

Latent Class		
Variable	Class 1	Class 2
Patterns	Esthetic pattern	Functional pattern
**SOCIODEMOGRAPHIC AT INCLUSION**		
**Age** (**ref <50**)		**0.98** [**0.96–0.99**]
≥50	1	**0.68 [0.49–0.95]**
**Education** (**ref ≥ High school**)		
<High school	1	**0.72** [**0.53–0.98**]
**Marital status** (**ref = Single**)		
Married/partners	1	**1.42** [**1.01–1.98**]
**Occupational status** (**ref = managerial occupations**)		
Performing trades (farmers, craftsmen, labourers, employees)	1	0.74 [0.51–1.07]
Unemployment at the time of diagnosis	1	0.39 [0.26–1.59]
**LIFESTYLE AT VICAN2**		
**BMI** (**kg/m^2^**) (**ref = normal**)		1.01 [0.98-1.03]
Underweight (<18.5 kg/m^2^)	1	1.91 [0.85-4.32]
Overweight (25–30 kg/m^2^)	1	1.20 [0.83–1.72]
Obese (>30 kg/m^2^)	1	1.26 [0.77–2.06]
**Physical activity** (**ref = stable**)		
Stop	1	1.74 [0.93–3.28]
Decrease	1	**1.77** [**1.23–2.56**]
Increase	1	1.15 [0.74–1.82]
**Smoking** (**Ref: No**)	1	1.24 [0.84–1.83]
**Alcohol consumption** (**Ref: No**)	1	0.84 [0.59–1.20]
**TREATMENTS AT VICAN5**		
**Chemotherapy**		
Yes	1	**1.63** [**1.19–2.24**]
**Radiotherapy**		
Yes	1	1.31 [0.88–1.94]
**Hormone therapy** (**ref = no**)		
Yes, and I’m still taking this treatment	1	**1.53 [1.07–2.19]**
Yes, and I’m no longer taking this treatment	1	1.06 [0.71–1.60]
**Reconstructive surgery**		
Yes	1	1.14 [0.80–1.63]
**Pain score** (mean)	1	**1.14** [**1.04–1.26**]
**Fatigue score** (mean)	1	**1.02** [**1.01–1.03**]
**Comorbidities score** (mean)	1	1.57 [0.99–2.47]

## Data Availability

Data available on request due to restrictions. The data presented in this study are available on request from ADB. The data are not publicly available due to the privacy and ethical reasons.

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
