# Peer review of "Patterns of Sequelae in Women with a History of Localized Breast Cancer: Results from the French VICAN Survey"

_cancers, 2021, doi:10.3390/cancers13051161_

Round 1

Reviewer 1 Report

The opening sentence in the Simple Summary does not communicate well.

Lacking a citation for the opening sentence of the paper and please use current data rather than 2018?

In the study population first sentence, add French female adults, rather than leaving to the end of the section. Men can be included in BC studies and sex should be clarified as early as possible.

Define esthetic. This is not a universal category and has little meaning to this reviewer. Where is it reported in Table 2?

2.2.1. Data on spontaneous sequelae. There is a sentence here that needs clarification. Do the authors mean the qualitative data were reviewed and grouped into themes? This would be a strange process for questionnaire items that are most likely already in tested subscales. Please add more clarification to this section. Perhaps separating the qualitative and quantitative data explanations would take care of this confusion.

2.2.3. Medical characteristics. This list of types of data does not match the list in the introduction. Please be consistent.

Add validity and reliability for all standardized measures. Add range of anchors and total scores. There is inadequate information on the measures.

2.2.4. Comorbidity. Add a sentence as to why there is no reliability for this scale. If it is due to being a binary scale, say so.

3.1. Participants’ characteristics. No summary statement on comorbidity in text, yet Table 1 starts with comorbids.

Figure 2:  A key or text is needed for Figure 2. It is not self-explanatory.

3.3. Latent-class analysis of sequelae patterns: provide rationale for collapsing categories. Again, what is esthetic? Also, add the time point to the Table and text. Do not assume VICAN5 is clearly year 5 for all readers.

Table 3 needs more explanation. How do the two groups across the top relate to the identical named groupings on the vertical left? Is this how the authors derived the Class 1 and 2? If so, for example why does Class 1 not include psychological in the collapsed definition above? Clarify.

4.1. Principal findings. 4.1. Again, reconstruction is omitted from the list for Discussion.

Clarify what this study adds to the science above and beyond Schmitz KH, Speck RM, Rye SA, DiSipio T, Hayes SC. Prevalence of breast cancer treatment sequelae over 6 508 years of follow-up: The Pulling Through Study. Cancer 2012;118:2217–25. 509 https://doi.org/10.1002/cncr.27474.

4.3. Supportive care for patients with cancer: supportive care is both promoted and seen as something to reduce in the same paragraph. Clarify which it is.

“Only one systemic review in breast cancer patients receiving hormone therapy concluded that aerobic plus resistance training had positive effects on cardiorespiratory fitness and pain, but further studies are needed to confirm the associations.”  Add the citation.

This highlighted statement in the Conclusions is an overstatement of findings from this study: “prevention measures through lifestyle changes with physical activity could prevent future sequelae.”   There was only an association found at year 5 between decrease in activity and sequela. These data are inadequate to recommend physical activity. A confirmatory RCT might be in order, but this study was not prescriptive.

Also, in limitations it is stated:  “changes in physical activity levels of patients since diagnosis were not measured by validated questionnaires or by objective tools such as accelerometers to objectively quantify their physical activity level.”  Inadequate measurement further clarifies that more definitive work is needed prior to a clinical recommendation.

Reviewer 2 Report

The article is innovative and well written. I do no have any major remarks except the fact that I would evaluate breast cancer in young women separately (women younger than 40 as it is a specific category of patients - https://doi.org/10.3390/cancers11111791.)

Reviewer 3 Report

“Patterns of sequelae in women with a history of localized breast cancer: results from the French VICAN survey”

The study has identified important subgroups of sequelae and associated risk factors at five years since diagnosis in patients with early breast cancer using the latent class analysis model. Two main classes were identified functional and esthetic sequelae.

I have an important concern regarding how were measured the different sequalae while no valid instrument were used for most of them. 

More information is needed in how many BC patients were available in Vican 2 and Vican 5 separately before applying the study selection criterias. This will allow to appreciate the attrition rate between the two time points two and five. Further, more information is needed about the overall response rate and eligibility criteria in Vican.

In relation with Fatigue, it looks like only fatigue corresponding questions from the EORTC QLQC30 were used? If it the case, then how did the authors guarantee the validity of the tools and that the psychometric properties of the scale were respected? If BC cancer patients went through the whole QLQC30 questionnaire (this need to be clarified in the text) then why the authors do not use all of sequelae measured with the validated and well-known instrument? 

There is no information on missing on the covariates and how the authors handled it. Only marital status had missing values (only 1?). There was no missing information in employment status and other variables?  

In the distribution of the 6 main sequelae figure 2, sequelae-free represents 28% at Vican2 and 31% at Vican5, these rates are as much important as functional sequelae and more important than esthetic sequelae, it is therefore strange that only two classes were retained from latent class analysis. Having a third sequelae-free class  would have been more natural. This sequelae-free class may have later been used as reference group in the logistic regressions.  

Round 2

Reviewer 3 Report

No other comments